# CRISPR-Based Genome Editing Tools: An Accelerator in Crop Breeding for a Changing Future

**DOI:** 10.3390/ijms24108623

**Published:** 2023-05-11

**Authors:** Fangning Zhang, Ting Xiang Neik, William J. W. Thomas, Jacqueline Batley

**Affiliations:** 1College of Life Sciences, Shandong Normal University, Jinan 250014, China; 2School of Biosciences, University of Nottingham Malaysia, Semenyih 43500, Malaysia; 3School of Biological Sciences, University of Western Australia, Perth, WA 6009, Australia; 4School of Biological Sciences, Institute of Agriculture, University of Western Australia, Perth, WA 6009, Australia

**Keywords:** plant genome editing, CRISPR-Cas systems, crop breeding

## Abstract

Genome editing is an important strategy to maintain global food security and achieve sustainable agricultural development. Among all genome editing tools, CRISPR-Cas is currently the most prevalent and offers the most promise. In this review, we summarize the development of CRISPR-Cas systems, outline their classification and distinctive features, delineate their natural mechanisms in plant genome editing and exemplify the applications in plant research. Both classical and recently discovered CRISPR-Cas systems are included, detailing the class, type, structures and functions of each. We conclude by highlighting the challenges that come with CRISPR-Cas and offer suggestions on how to tackle them. We believe the gene editing toolbox will be greatly enriched, providing new avenues for a more efficient and precise breeding of climate-resilient crops.

## 1. Introduction

Plant breeding techniques have progressed rapidly since the introduction of biotechnology in the early 1980s, starting with the discovery of transgenics methods, and further developing into molecular marker systems and genomics approaches in the late 1980s to 90s for applications in crop improvement [1,2]. While genomics and marker-assisted selection have become the mainstream methods of plant breeding, with high success rates in delivering high-quality crops with increased yields over the past two decades, these genetic advances are limited by the extent of novel genetic variation in the crop gene pool [3], which is key for breeding new crop varieties. New plant breeding methods [4] including gene editing have come to the fore to bridge this gap, offering very promising breeding outcomes with high speed and precision through targeted mutagenesis to obtain new varieties with improved traits [5,6].

With the increasing amount of crop genome data available, genome editing is now widely used to improve crop quality and yield through genetic modifications targeted at the gene level [7]. There are several gene editing methods, all of which are adaptations from naturally occurring DNA break and repair mechanisms. The zinc finger protein, first discovered in *Xenopus*, contains repetitive zinc binding domains that grasp the DNA in a DNA–protein interaction [8], where the zinc finger is further exploited to fuse with specific DNA (or RNA) sequences that have a DNA-cutting function to form a zinc finger nuclease (ZFN) [9,10]. A similar approach is transcription activator-like effector nucleases (TALEN) involving a DNA–protein interaction; its discovery is from the effector protein secreted by the bacterium *Xanthomonas* that plays a role as a bacterial transcription factor regulating the susceptibility genes in host plants. The effector protein possesses a DNA-recognition domain and can be manipulated to link with an endonuclease (e.g., FokI) for genome editing [11,12]. ZFN and TALEN are ideal in recognizing specific DNA sequences and perform cleavage functions on double-stranded DNA. However, the main limitations of these techniques are the cumbersome process of constructing a new pair of ZFNs or TALENs each time one wants to edit multiple sites and the high tendency for off-target effects, which could lead to detrimental effects on the cell. Hence, the more recent CRISPR-Cas system has taken centre stage in the genome editing world.

CRISPR-Cas is a revolutionary gene-editing technology that allows researchers to make precise modifications to DNA sequences in living cells. CRISPR stands for “Clustered Regularly Interspaced Short Palindromic Repeats” and refers to a unique pattern of repeating DNA sequences found in bacteria and other microorganisms. Cas (CRISPR-associated) refers to the proteins that work alongside the CRISPR sequences to edit DNA.

In CRISPR-Cas technology, a specific RNA molecule is designed to bind to a specific target DNA sequence, and the Cas protein is used to cut the DNA at that location. CRISPR-Cas offers many advantages over ZFNs and TALENs. It is generally considered more versatile than ZFNs and TALENs, it is easier to design and implement with greater editing efficiency outcomes and it can be programmed for multiplexing [13,14]. Recent progress in CRISPR-Cas systems allow researchers to modify the gene function, expression and regulation at considerably high levels of specificity and accuracy at the single-base level using single-base and prime-editing approaches [15,16,17].

Base editing (BE) is an improvement to the CRISPR/Cas9 system to overcome the frequent unintended editing effects brought upon by double-stranded break (DSB) repair mechanisms such as indels, chromosomal translocations, rearrangements and tumorigenesis [18,19]. In base editing, the Cas9 nuclease activity is disabled and replaced with a nickase Cas9 (nCas9), which is fused with a base deaminase, thereby introducing single-strand breaks instead of DSBs. This is achieved using either the cytosine base editor (CBE, C converts to T, G converts to A; in other words, conversion of C:G to T:A base pair, an example of base transition) [20,21], or the adenine base editor (ABE, A converts to G, T converts to C, conversion of A:T to G:C base pair, another example of base transition) [22,23]. Base editing in cereal crops has been reported to offer great prospects in inducing efficient point mutations for improved agronomical traits [24]. For example, the enhancement of grain weight and crop yield in rice (*Oryza sativa*) [25], and herbicide tolerance in wheat (*Triticum* spp.) and maize (*Zea mays*) [26,27].

There have been numerous advances in the BE systems. A combination of CBEs and ABEs in a so-called “dual deaminase CRISPR base editor”, namely CRISPR-Cas9-based Synchronous Programmable Adenine and Cytosine Editor (SPACE) [28], Adenine and Cytosine Base Editor (ACBE) [29], Saturated Targeted Endogenous Mutagenesis Editors (STEMEs) [30] and Target-ACEmax [31], were developed to engineer the simultaneous conversion of A-to-G and C-to-T transitions at a target site. In rice using STEMEs, a near-saturated mutagenesis of 73.21% was obtained for the rice coenzyme A carboxylase (OsACC), conferring herbicide resistance. [30]. Continuous improvements to the efficiency of dual base editors in plants have been reported, thus expanding the CRISPR-Cas toolkit for plant genome editing [32].

Prime editing (PE) was designed to cover the limitations of base editors in that it performs all types of single and multiple base substitutions including transversion and transition types of point mutations, as well as insertions and deletions, which offers higher flexibility than CBEs and ABEs [33]. In crops, the initial application of PE was in rice and wheat [34], and it was further applied in maize to create mutations in two acetolactate synthase (ALS) genes, conferring a herbicide-resistance trait to the plant [35] with ongoing improvements in PE systems in plants for greater editing efficiency [36,37,38].

Herein, we delineate the different classes and types of CRISPR-Cas systems with their corresponding features, discuss the application of each of these systems in plant research and provide updates on emerging and promising genome editing systems that could benefit plant breeding research.

## 2. The Immune Response Stages and Classification of CRISPR-Cas Systems

CRISPR-Cas is an adaptive immune system found in bacteria and archaea that plays a role in defence mechanisms against invading pathogens, such as virulent phages and plasmids, and also offers protection against invasion of harmful foreign mobile genetic elements (MGEs) acquired through horizontal gene transfer [39]. The CRISPR-Cas system works by integrating fragments of foreign nucleic acids into CRISPR arrays as new spacers. These new spacers are located between pre-existing palindromic repeats that are typically 25–35 base pairs long and are separated by unique spacers that are usually 30–40 base pairs long. By acquiring new spacers into the CRISPR array, the immune memory is stored, and the system is able to recognize and target specific pathogens for destruction in future encounters [40,41].

There are three stages in the CRISPR-Cas immune response: adaptation, expression and processing of the precursor CRISPR RNA (pre-crRNA) and interference [42,43], which are the basic mechanisms for the systems to carry out programmable genome editing. Detailed information of the three steps is described below (Figure 1).

Adaptation stage. The CRISPR-Cas system employs a complex of Cas proteins that bind to a target DNA by recognizing a specific protospacer-adjacent motif (PAM), a short DNA sequence located immediately adjacent to the target sequence. This recognition triggers the production of double-stranded breaks (DSBs) in the target DNA with a segment excised. This excised segment, called the protospacer, is then inserted between two repeats of the CRISPR array, where it becomes a spacer. During this adaptation stage, a complex of Cas1 and Cas2 proteins serves as the adaptation module, which is a highly conserved structure present in most CRISPR-Cas systems. Cas1 acts as an endonuclease to cleave both the target DNA and CRISPR array, while Cas2 forms the structural scaffold of this complex [44,45]. Additional proteins, such as Cas4, can fuse with Cas1 or Cas2 to contribute to the adaptation stage [46]. In some Cas13 subtypes, the *cas1* and *cas2* genes are absent from their CRISPR-Cas loci but can be obtained from other CRISPR-Cas systems within the same genome. This allows for the functionality of Cas1 and Cas2 to be provided from an additional Cas locus of either type I or type II [47].Pre-crRNA expression and processing stage. The CRISPR array is transcribed into pre-crRNA, which is then processed into mature crRNAs through different mechanisms depending on the CRISPR-Cas system. For example, in most class 1 CRISPR-Cas systems, a dedicated processing nuclease called Cas6 accomplishes the processing, while in class 2 CRISPR-Cas systems, a housekeeping RNase III is responsible for processing in the case of CRISPR-Cas9. In other class 2 systems, crRNA processing is carried out by a single large Cas protein [42,48,49].Interference stage. After the pre-crRNA is processed into mature crRNAs, it can form a complex with a trans-acting crRNA (tracrRNA), and together they act as a guide RNA (gRNA). The effector module, either a Cas protein complex in class 1 or a single large Cas protein in class 2, is then directed by the gRNA to recognize the specific PAMs in target DNA or RNA. Upon binding, the Cas nuclease is activated when the crRNA spacer is base-paired with the complementary target strand, resulting in the cleavage of the target DNA/RNA [48].

Encoded by *cas* genes associated with the adjacent CRISPR array, Cas proteins are a cluster of multiple endonucleases, and major components of both adaptation and effector modules as described above [50]. The constituents of the adaptation module are conserved, while the components of the effector module vary between groups. This variation comes from the co-evolution during the perennial arms race between mobile genetic element (MGE) and host CRISPR-Cas systems, which underlies the classification of modern CRISPR-Cas systems.

There are two distinct classes of CRISPR-Cas systems based on the makeup of the effector module [51]. In class 1, the effector module comprises a multiple-subunit complex containing several Cas proteins. In class 2, the effector module consists of a single, large and multidomain Cas protein (Figure 1B,C), the structure of which is always bilobed, containing the REC (nucleic acid recognition) lobe and NUC (nucleic acid cleavage) lobe [52]. Further differences are also recognized in how these nucleases conduct processing. In class 1, the CRISPR-associated complex for antiviral defence (Cascade), comprising multiple Cas proteins, binds to the pre-crRNA and Cas6 is employed for processing (Figure 1B) [53,54]. In class 2, instead of Cas6, the tracrRNA plays a critical role in processing, particularly in type II and type V-B of the CRISPR-Cas systems [51,55,56]. Other major differences pertain to the nuclease domain for DNA cleavage. In class 2, a single large Cas protein harbors the nuclease domain for cleavage (Figure 1C). In class 1, the nucleases are more diversified between types and subtypes. There are different types and subtypes in class 1 and 2—type I, III and IV in class 1 (multi-unit effector molecules) and type II, V and VI in class 2 (single effector molecule). Each type is defined by its effector module, especially the subunit for target cleavage (Table 1).

## 3. CRISPR-Cas Types and Their Application in Plant Genome Editing

After the target breaks are introduced, cellular endogenous DNA repair pathways are carried out by non-homologous end joining (NHEJ), microhomology-mediated end joining (MMEJ) or homology-directed repair (HDR) [57]. NHEJ is the main repair pathway, resulting in editing at the target sites and introducing favourable modifications, but it is error-prone and often generates indels in the repair process [57,58,59].

Currently, Cas9 is the most prevalent Cas effector used in genome editing; however, other nucleases, such as DNA-targeting Cas12a/Cpf1 and RNA-targeting Cas13, have received considerable attention. There are also emerging Cas proteins and systems which show advantageous features, possibly overcoming some of the current drawbacks of CRISPR-Cas systems and pushing the boundaries of genome editing [60]. Here, the characteristics, specific domains, applications and limitations of each CRISPR-Cas type are discussed within the context of plant genome editing, starting with well-known types and then those that are emerging.

### 3.1. Class 2 Type II CRISPR-Cas9 Systems

The class 2 type II dual-RNA-guided DNA-targeting endonuclease Cas9 was the first Cas protein exploited for genome editing, among which SpCas9 (Cas9 from *Streptococcus pyogenes*) is the most commonly used [61]. In CRISPR-Cas9 systems, a gRNA is formed by annealing the mature CRISPR RNA (crRNA) to its corresponding tracrRNA. The resulting gRNA directs the Cas9 to bind to the target DNA, ultimately resulting in DSBs. As a multifunctional effector, Cas9 has two distinct nuclease domains in its NUC lobe, HNH and RuvC-like, which cleave the target and non-target strands, respectively, and the nucleases of both domains are activated by correct base pairing [42,62]. The recognition of the target strand is dependent on G-rich PAMs, which are canonical NGGs (N is A, T, C, or G) in the case of SpCas9. Blunt ends created by HNH and RuvC-like domains are proximal to the PAM (Figure 2A) [61]. Besides Cas1 and Cas2, Cas4 also takes part in the adaptation stage in subtype II-B by binding to Cas1 [63]. In the pre-crRNA processing stage, tracrRNA and the conserved endonuclease RNase III process the pre-crRNA into mature crRNA [55,56] (Figure 1C). In the interference stage, tracrRNA and crRNA trigger site-specific DNA cleavage by the two domains in Cas9. Cas9 orthologues acquired from different bacterial strains show sequence variability in PAM and Cas9 sizes, suggesting potential to expand the genome modification sites.

In 2013, CRISPR-Cas9 was applied in HDR-mediated genome modification in rice protoplasts [64]. Strategies vary in its application for precise plant genome modification, such as knockout of genes expressing undesirable traits, and knock-ins or replacements of new alleles encoding anticipated phenotypes, with applications in yield and quality improvements, and stress tolerance as summarized by Chen et al. [65]. For example, completely sterile *spo11-1* mutants were generated in hexaploid wheat for the first time, using CRISPR-Cas9 to edit all three homoeologues of *SPO11-1* in a loss-of-function manner [66]. This study not only revealed the role of *SPO11-1* in meiosis, but also supported the possibility of editing meiotic recombination in wheat using CRISPR-Cas9 systems, potentially increasing the efficiency of plant breeding programs by reducing the time and resources required to produce crops with improved nutritional content, increased yield or other desirable traits. Another remarkable feature of the CRISPR-Cas9 system is the RNA-targeting SpCas9 (RCas9). Using synthetic PAM-presenting oligonucleotide (PAMmer), RCas9 is able to recognize and cleave RNA in a programmable manner. Given a matching gRNA, the PAMmer allows precise manipulation of RNA molecules, thereby expanding the scope of plant genome editing applications [67]. The inactivation of either the HNH or RuvC-like domain creates a Cas9 nickase (nCas9) that is capable of cleaving only one DNA strand. This property makes nCas9 useful in base editors and prime editors [68]. This base editing technique has been shown to effectively modify the *Pi-d2* gene in rice, a type of R gene that has high GC content, and in which a non-synonymous mutation causes the plant to be susceptible to *Magnaporthe oryzae* [69]. The inactivation of both nuclease domains generates catalytically inactive Cas9 (dCas9), which still binds to target DNA but lacks any nuclease activity, giving rise to CRISPR interference (CRISPRi) systems, or CRISPR activation (CRISPRa) systems when fused with transactivator domains [70,71]. The effectors employed by CRISPRa/CRISPRi systems to specific sites have been used for regulating gene expression, modifying the epigenome and visualizing genomic loci or telomeric regions [72,73,74,75]. Understanding the dynamics of chromosomes, including DNA–protein interactions and studying the 3D chromosomal conformations, potentially through CRISPR-based live cell imaging, could provide us valuable insights into the evolutionary history of domesticated traits in complex polyploid plant genomes. In *Arabidopsis*, CRISPRa/CRISPRi was applied to improve the plant’s tolerance towards drought stress [76,77,78]. However, some disadvantages have been found in these derivative editing tools. For example, the catalytic domains of the dCas9-effector may become fused to neighbouring or even unrelated loci, resulting in off-target effects. Additionally, the efficacy of these systems could be species-dependent, which could limit their further promotion [79].

Despite the wide application and significant revolution in biological sciences using CRISPR-Cas9 systems, there are obvious drawbacks, such as its off-target effect, dependence on specific PAM sequences and large gene size. To fully leverage the potential of the CRISPR-Cas systems, genome editing tools developed from other types have been applied and assessed using CRISPR-Cas9 as a benchmark, which will be summarized below.

### 3.2. Class 2 Type V CRISPR-Cas12a (Cpf1) Systems

Compared to type II (Cas9), type V-A Cas12a (Cpf1, from the bacteria *Prevotella* and *Francisella*) is smaller in size and contains two RuvC-like domains in its NUC lobe, which is superimposed once activated by correct base pairing, cleaving both target and non-target strands and generating cohesive ends [80,81] (Figure 2B). Unlike Cas9, the PAM site of Cas12a is T (thymidine)-rich, such as TTTV (V is A, C, or G), and DNA cleavage is distal from the PAM, generating 5′ staggered DSBs [80,82]. Cas12a was the first Cas12 nuclease used for genome editing, with dual functions not only in processing pre-crRNA into mature crRNA without the assistance of tracrRNA (that is cleaving its own crRNA), but at the same time cleaving the target sequence, suggesting Cas12a is useful for multiplexed genome editing [83].

Cas12a has been widely used in mediating plant genome modification, enabling gene deletion, insertion, base editing or locus tagging in rice and other economically important plants such as macroalgae and citrus [83,84,85]. CRISPR-Cpf1 and CRISPR-Cas9 were comparatively studied in rice, mediating the knockout editing of the *EPFL9* (*Epidermal Patterning Factor like-9*) gene. Accordingly, the CRISPR/LbCpf1 (LbCas12a) system generated a higher mutation percentage and longer deletion size than that of Cas9 [86]. A system named STU (single transcript unit)-Cas12a has been developed to facilitate both single and multiplexed genome editing in rice [87]. In both transient expression and stable transgenic T0 lines, the transformation of four single STU-Cas12a systems targeting one crRNA each, and one multiplexed STU-Cas12a system targeting an array with four crRNAs, was carried out, achieving considerable editing efficiency (29.2% to 50%) at the four independent target sites [87]. This suggests that the CRISPR-Cas12a system can be used to efficiently edit multiple target sites and generate heritable mutations in plant genomes, which could lead to the production of crops with improved yields, resistance to pests and diseases and other desirable traits. Furthermore, the catalytically dead Cpf1 has been reported to function as a transcriptional repressor in plants and bacteria, demonstrating a more than 10-fold reduction in *mi*R159b transcription in *Arabidopsis*, suggesting its promising application in plant transcriptome regulation [88]. This represents a powerful tool for gene silencing in gene functional studies. It enables us to investigate the regulation and expression of target genes, and to establish connections with complex phenotypes, such as quantitative resistance in plants.

### 3.3. Class 2 Type VI CRSIPR-Cas13 Systems

Type VI CRISPR-Cas13 (C2c2) is the only known system to exclusively bind and cleave single-stranded RNA (ssRNA) among the six major CRISPR-Cas systems. Cas13 is a kind of RNA-guided RNA-targeting ribonuclease. The mechanisms of protospacer acquisition differ between class 2 subtypes, with or without the adaptation module, Cas1 or Cas2, in the adaptation stage [51]. The processing stage of the CRISPR-Cas13 system is mainly carried out in the REC lobe, where Cas13 binds to the pre-crRNA and cleaves within the crRNA direct repeat (DR) to generate a mature crRNA. Unlike Cas9 or Cas12, which requires a PAM to recognize the target strand, some Cas13 proteins have a preference for protospacer flanking site (PFS) [52,89,90] (Figure 2C). All known Cas13 nucleases contain two distinct HEPN (higher eukaryote and prokaryote nucleotide binding) domains in the NUC lobe, which are recognized as the active sites for targeted RNA cleavage. In the interference stage, the Cas13-crRNA complex binds to the targeted ssRNA, a ternary ribonucleoprotein (RNP) complex is then formed, following the precise complementary pairing between crRNA seed region and target RNA [91,92]. A single catalytic site is generated following the conformational reordering and spatial proximity of two HEPN domains after the formation of an RNP complex, where the ssRNA is cleaved [52,90,93].

Owing to its RNA-targeting idiosyncrasy, Cas13 has immense potential in applications in plant research, such as targeted RNA knockdown, RNA virus defence and epitranscriptome modification [94]. The knockdown efficiency of LwaCas13a (Cas13a from *Leptotrichia wadei*) was tested in rice protoplasts, with three guides designed for each of the three endogenous transcripts, resulting in >50% knockdown for seven out of the nine gRNAs, with no collateral degradation detected [91]. This result exhibits the successful manipulation of cytoplasmic RNA in plants via Cas13a with satisfying efficiency, which implies a significant achievement for crop breeding and improvement, given the possibility to manipulate coding RNAs, such as mRNAs, and non-coding RNAs, such as microRNAs (miRNAs) and long non-coding RNAs (lncRNAs), which regulate and control important plant characteristics. Furthermore, the authors identified that dCas13a can be used as a programmable RNA-binding protein in mammalian cells, which is useful in tracking transcripts in live cells, similar to RCas9 [91]. LshCas13a (Cas13a from *Leptotrichia shahii*) has been tested to engineer immunity against RNA viruses, such as tobacco turnip mosaic RNA virus (TuMV) and potato virus Y (PVY), in both monocot and dicot plants, presenting moderate efficiency [95,96,97,98,99], offering a promising opportunity to breed disease-resistant plants using CRISPR. In their study, Mahas et al. [100] investigated nine different Cas13 variants and found that CasRx (Cas13d ortholog from *Ruminococcus flavefaciens*) exhibited the highest efficiency in RNA virus targeting and interference in tobacco (*Nicotiana benthamiana*), without producing any collateral cleavage activity. These findings support a highly efficient and precise genome editing in plants, particularly in the search for editing elements sourced from bacteria. A Cas13d subtype, which is smaller in size than Cas13a, has been discovered to function effectively across a broad temperature range (i.e., 24–41 °C), in some cases making it well-suited for use in highly sensitive reverse transcription recombinase polymerase amplification (RT-RPA) for nucleic acid detection. This feature makes Cas13 enzymes an ideal candidate for developing molecular diagnostic tools [101,102,103], for example developing a CRISPR-based diagnostic kit for the early detection of plant pathogens in the field [104]. Furthermore, Cas13d targets RNA molecules without a strong preference for a specific PFS sequence but rather recognizes the uracil base within the target RNA, which enables Cas13d to target a wider range of RNA molecules. It also tends to degrade unfolded ssRNA while avoiding RNA with complex secondary structures [105]. In addition, dCas13d is generated by inactivating the HEPN domains, thereby retaining its target-RNA-binding capacity while losing the cleavage activity. The dCas13d can then be fused with a modified plant APEX2 (ascorbate peroxidase) to enable the detection of RNA–protein interaction in vitro [106]. All these properties make Cas13d a powerful editing scissor in transcriptome engineering.

Another cutting-edge application of dCas13 in plant research is the programmable m^6^A (N6-methyladenosine) modification. m^6^A is the most prevalent and abundant post-transcriptional mRNA modification in plants, which plays an important role in regulating plant growth and development as well as biotic/abiotic stress resistance [107]. The development of CRISPR-Cas13 systems provides a convenient tool for m^6^A modification via targeting mRNA with higher efficiency and precision [108]. Two steps are included in the m^6^A editing process: (1) dCas13 bound to the target RNA specific site directed by gRNA and PFS at the single-base resolution and (2) writer (methyltransferase, such as METTL, MTA or FIP) or eraser (demethylase, such as ALKBH) enzymes fused to dCas13 to add or remove m^6^A at the target site. Reader enzymes (m^6^A binding proteins, such as ECT) are recommended to be added to the dCas13-eraser complex to decrease the off-target odds [108,109].

A recent noteworthy discovery shows that Cas13-guiding crRNA can lead to significant reductions in RNA levels even in the absence of the Cas13 protein, as observed in *Arabidopsis*, tobacco and tomato (*Solanum lycopersicum*) species [110]. Cas13-independent guide-induced gene silencing (GIGS) is speculated to be functional in numerous eukaryotes, and it holds great promise as a compact, multigene silencing editing tool. This can be particularly useful in polyploid plants, where there are multiple copies of a specific gene or genes in the genome and where it is necessary to modify multiple genes simultaneously to achieve desired quantitative traits such as increased yields and disease resistance. Further studies could be carried out to take full advantage of this novel gene editing approach in crop improvement.

### 3.4. Class 1 Type I Systems

Class 1 type I CRISPR-Cas systems are the most abundant CRISPR-Cas systems among bacteria and archaea. Cas3, the signature protein of type I, has a histidine-aspartate (HD) nuclease, functioning as a helicase-nuclease [111,112]. While Cascade-Cas3 has been widely used in prokaryotic genome manipulation, its use in eukaryotes was impeded by the multiple-subunit effectors, which required the simultaneous or sequential introduction of multiple genes, until the recent creation of the repurposed Cascade-Cas3 [113,114]. The long target sequence (about 30 bp) recognized by type I CRISPR-Cas systems offers great advantages to genome editing [115].

The structure of type I-E has been well studied, in which Cascade was first identified [54,116] (Figure 2D). In the type I-E CRISPR-Cas system, the Cascade complex comprises five Cas proteins: Cas5e, Cas6e, Cas7e, Cas8e and Cas11e and crRNA. Upon PAM recognition by Cas8, an R-loop structure is formed between crRNA and the target DNA. Subsequently, a specific nuclease Cas3 is recruited, and the target DNA is cleaved and degraded [111,112,117,118,119,120].

The only known application of type I-E in plant cells is the transcriptional control in maize (*Zea mays* L.), using the type I-E system from *Streptococcus thermophilus* [115]. Using this system, the anthocyanin biosynthesis gene was successfully activated with an approximately twofold enhancement and resulted in red pigmentation at the aleurone cell layer in *Zea mays* [115]. In type I-E CRISPR, the presence of multiple activation domains bound to the Cascade subunits at target sites may result in a more consistent and enhanced activation of multiple genes, making it a suitable tool for investigating gene regulatory networks and how genes are controlled, as compared to a single effector.

A recently discovered type I-D CRISPR-Cas system, TiD, originally found in the bacteria *Microcystis aeruginosa*, has been adapted as a genomic editing tool in eukaryotes [121,122]. TiD consists of five Cas proteins, Cas3d, Cas5d, Cas6d, Cas7d and Cas10d, and a crRNA (Figure 2D). One unique feature of TiD is that it contains a mixture of type I and type III effector modules—the helicase in type I (Cas3′) and Cas10d in type III fused with the HD nuclease domain (Cas3″) to perform cleavage and ssDNA nuclease activity in vivo [121]. This hybrid nature of effector proteins demonstrating dsDNA and ssDNA cleavage qualifies this system, TiD, as a powerful tool in editing the inaccessible regions of complicated crop genomes, for example in *Brassica*, wheat, cassava and potato, for trait improvements such as disease resistance and high nutritional content [123,124]. Furthermore, TiD lacks the Cas8 homologous protein that is responsible for PAM recognition. Instead, Cas10d is involved in recognizing PAMs with the sequence 5′-GTH-3′ (H is A, C or T) [46,121,125,126]. Non-canonical small subunits such as Cas11d, which is a protein produced from within the Cas10d gene (“hidden” components of the CRISPR system), are the first known example of alternative internal translation, where the same mRNA transcript can produce multiple proteins by initiating translation at different internal sites [126,127]. This finding is significant because it expands our understanding of how genes can produce multiple proteins and how these proteins may interact with each other in cellular processes. The other unique feature of TiD is the capacity to perform small indels and long bi-directional deletions (up to 7.2 kb) as shown in tomato in transgenic calli and shoots, targeting parthenocarpy and fruit ripening genes [60]. More on-target sites for TiD were found in tomato and *Arabidopsis* compared to Cas9, and these on-target mutations can be transmitted to the next generation [121]. Given that Cas10 is regarded as the hallmark protein of the type III CRISPR-Cas system, and given the similar structure of crRNA and Cas7 in type I-D to that of type III, it is possible that type I-D represents an evolutionary intermediate between type I and type III CRISPR-Cas systems [127]. The understanding of this hybrid nature and its distinct recognition to dsDNA/ssDNA, especially the characteristics of the specific Cas10 protein, can help researchers identify new components and mechanisms of these CRISPR systems, and potentially improve the efficiency and specificity of genome editing tools for crop improvements [126].

Besides Cas3, Cas11 is another key effector in type I systems, which shows diversity across the type I subtypes. In type I-E systems, Cas11e subtypes are encoded by a well-annotated, independent *Cas11* ORF; in both type I-C and type I-D systems, Cas11 subunits are “hidden” components encoded from domains in Cas8c and Cas10d, respectively [127,128,129]. Cas11 effectors serve not only as essential components in most of the type I systems [129], but also contribute to programmable RNA knockdown when fused with Cas7 in type III systems [130]. Considering the numerous distinctive characteristics of class I CRISPR-Cas systems commonly present in complex prokaryotic genomes, this system holds great potential and is advantageous for editing the plant’s DNA to enhance its traits.

### 3.5. Class 2 Type V-B CRISPR-Cas12b/C2c1 Systems

Cas12b/C2c1 features a conserved Ruvc-like domain and a putative NUC domain that differ entirely from those of Cas12a [131]. As a type V system, CRISPR-Cas12b/C2c1 uses a chimeric gRNA formed by hybridization between crRNA and tracrRNA to guide its endonuclease activity [51,132] (Figure 2E). It recognizes T-rich PAMs and produces a long-staggered end distal to the PAM, resulting in 5′ overhangs. The unique feature of Cas12b includes the recognition of a T-rich PAM sequence instead of G-rich in Cas9, which means a wider range of targeting sequence can be achieved. Furthermore, the size of Cas12b is smaller than Cas9, making it an efficient and convenient delivery tool for gene editing [133]. Cas12b is by far the only protein in CRISPR-Cas that produces the longest number of nucleotides (6–8) in the sticky end [134] compared to the 1–3 nucleotides overhang in Cas9 [135], which makes Cas12b useful because it minimizes editing errors during the NHEJ repair process. In addition, Cas12b’s ability to function in a wide range of temperatures and pH allows researchers to perform more efficient functional studies of resilient crops such as heat- and salinity-tolerant traits.

As reviewed by Wada, Osakabe and Osakabe [60], the editing efficiency of Cas12b proteins from various bacteria were studied in rice, among which AaCas12b (*Alicyclobacillus acidiphilus*) showed high mutation specificity. Additionally, AaCas12b demonstrated up to eightfold enhancement of gene transcriptional activation in rice, making it a very attractive CRISPR system for plant genome engineering [136]. Cas12b favours the PAM sequences of VTTV (V is A, C or G), leading to a high mutation efficiency (>50%) at ATTA, ATTC and GTTC PAMs in rice protoplasts in a target-site-dependent manner, with deletions occurring approximately 12–24 bp distal to the PAM sequence. Additionally, inactivated AaCas12b proteins containing transcriptional repression or activation domains can be used to regulate the expression of target genes [136]. AaCas12b has been demonstrated to function effectively at high temperatures, as demonstrated in cotton (*Gossypium hirsutum*) where the highest genome editing efficiency at 17.1% was observed at 45 °C for 4 days, with 1–16 bp deletions induced [134]; this makes AaCas12b a promising candidate for use in developing heat-tolerant crops. Other types of Cas12b, such as BvCas12b (*Bacillus* sp. *V3-13*) and BhCas12b v4 (*Bacillus hisashii*) in *Arabidopsis*, demonstrated possibilities of multiplexed genome editing and generating heritable mutations [137]. These advantages are important in stacking multiple transgenes at specific loci to ensure multiple traits can be genetically inherited for cultivar improvement and development [138]. The advantage of Cas12b lies in its ability to generate longer gene deletions in plants, making it possible to target and remove gene loci that control undesirable traits or disrupt large genetic elements, such as gene clusters or large cis-regulatory domains that control expression of desirable genes [139], with minimal off-target effects and greater precision and efficiency [132,140]. Although CRISPR-Cas12b has shown promise in plant cell editing, the low editing efficiency and survival rate observed in pear calli highlight the need for the development of temperature-insensitive systems before widespread implementation is possible [141].

### 3.6. Class 2 Type V CRISPR-CasΦ Systems

CRISPR-CasΦ, isolated from huge bacteriophage genomes, is a hypercompact genome editing tool with fewer spacers in its CRISPR array that lacks the CRISPR spacer acquisition machinery (Cas1, Cas2 and Cas4 proteins) [142] (Figure 2F). CasΦ (Cas12j) contains a C-terminal Ruvc-like domain, and it has a relatively low sequence similarity with other Cas proteins in type V CRISPR-Cas systems (Figure 2F) [143]. CasΦ utilizes a single RuvC-like active site that is dependent on magnesium (Mg^2+^) for both crRNA processing and target DNA cleavage, functioning on both dsDNA and ssDNA in cis [142,144]. This concentration of structural and functional elements results in the generation of compact Cas12 proteins. By recognizing minimal T-rich PAM sequences such as the 5′-TBN-3′ PAM (B is G, T, or C), Cas12 proteins are able to expand their target recognition capability in comparison to other Cas proteins [142].

One important advantage of CRISPR-CasΦ is the small size of the CasΦ protein, which has a molecular weight half that of the Cas9 and Cas12a proteins. Its size is approximately 70 to 80 kDa, which makes it suitable for packaging in a virus-based vector, allowing for the easy and efficient expression of transgenes [142].

In plant genome editing research, the minimal requirement of PAM using CasΦ-2 as shown in the *Arabidopsis* 8–10 bp deletion of the *PDS3* (phytoene desaturase 3) gene suggests a broader editing scope, where there is more flexibility in the target site selection, thereby increasing the chance of generating homozygous or biallelic mutations in plants [142]. Further optimization of the CRISPR-CasΦ-2 in *Arabidopsis* and tobacco shows promising results in terms of improved genome editing efficiency and specificity, and can be applied to economically important crops for enhancing their traits [145].

### 3.7. Other Types of CRISPR-Cas System

In type III systems, Cas10 is the signature protein containing two RNA recognition motif (RRM) domains, which is significantly different from type I-D [146,147]. The majority of type III encoded proteins contain CARF (CRISPR-Associated Rossmann Fold) and/or HEPN domains, with the former a predicted nucleotide-binding domain and the latter a ribonuclease, similar to that of Cas13. Some other type III encoded proteins contain CARF- and DNA-binding domains, presenting their flexibility in target recognition [148]. Among the multiple effectors, Cas7 is responsible for RNA cleavage, while Cas10 induces non-specific ssDNA nicks [149,150]. In type III-E, a single multi-domain effector called Cas7-11 (gRAMP) is identified to cleave RNA, which can be developed for RNA-targeting engineering [130]. Type III CRISPR-Cas systems have been utilized in virus detection and other prokaryote studies, with their wider application in mammalians and plants currently hampered by the system’s complex structure, which therefore makes it challenging to engineer and optimize for specific applications.

Type IV is comparatively less studied, currently limited to prokaryotes, with characteristics distinct from other CRISPR types. It is speculated to have evolved from type III, with a lack of various components, such as the absence of CRISPR array, a loss of small units such as Cas11, the partial deterioration of the large subunits such as Cas10 or Csf1 and a loss of target cleavage nuclease as well as a lack of Cas6 and even adaptation modules, having been detected among the subtypes [151]. The mechanism details and its potential application remain unclear. A recent study described that the innate *Pseudomonas oleovorans* type IV-A system targets DNA upon PAM recognition, and causes DNA interference with the absence of cleavage activity [152].

There have been diversified subtypes discovered in type V, besides the three Cas proteins mentioned above. Cas12e (CasX) has been adopted as a genome editing tool in eukaryote and human cells, with a target strand loading (TSL) domain facilitating target DNA strand cleavage [153]. Other various type V subtypes show RNA-guided nuclease in prokaryotic cells, with further study in eukaryotic organisms anticipated, reviewed by Liu et al. [154]. A brief summary of the advantages, limitations and suggested solutions of the various CRISPR/Cas systems is demonstrated in Table 2.

## 4. Further Applications in Plant Science

### 4.1. De Novo Domestication

Plant domestication has been essential in producing the elite crop varieties that are cultivated worldwide [159]. However, such intense selection, often for improved palatability or higher yields, has reduced the genetic diversity of cultivars, making them vulnerable to both biotic and abiotic stresses [160]. Crop wild relatives (CWRs) represent a largely untapped pool of beneficial traits that can be applied toward breeding more climate-resilient and durable cultivars [161]. Through recent advances in CRISPR-Cas9 editing, a novel pathway to domesticate CWRs in extremely short time frames has emerged. De novo domestication describes the editing of genes responsible for key agronomic traits in a crop wild relative so that it becomes high-yielding, while also having beneficial wild-derived attributes, for example improved nutrition, abiotic stress tolerance or disease resistance [162]. Owing to the rapid identification and improved understanding of genes associated with domestication and yield-related traits in many major crops, targeted gene editing of such genes is becoming more widespread, opening the door for de novo domestication to become a powerful approach to capitalize on the genetic diversity retained in CWRs [163] (Figure 3A).

The first example of de novo domestication was demonstrated in a wild tomato relative, *Solanum pimpinellifolium*, which naturally produces smaller fruit compared to cultivated tomato. Zsögön et al. [162] first identified a suite of domestication genes in tomato, and using a CRISPR-Cas9 multiplex editing approach established transgenic *S. pimpinellifolium* lines with loss-of-function alleles for four of these yield-related genes. Compared to the parental wild type, the T_1_ generation displayed ten times the number of fruits, which were three times as large [162]. In addition, the transgenic fruit contained five times the amount of lycopene, a carotenoid found in tomatoes with beneficial antioxidant properties, when compared to cultivated tomato [162]. Since this initial proof of concept, two other successful attempts to domesticate CWRs de novo using genome editing have been made. One example is the distant tomato relative, *Physalis pruinose*, or groundcherry, an orphan crop commonly grown in Central and South America [164]. Notably, Lemmon et al. [164] carried out de novo domestication with limited genomic resources as no groundcherry reference genomes were available. Instead, they generated whole-genome sequencing data and relied on the phylogenetic relationship between groundcherry and tomato to identify and edit orthologues of known tomato domestication genes using CRISPR-Cas9. As a result, improvements were made to plant architecture and yield, including a more compact growth form, and increased flower number and fruit size in the transgenic plants [164]. De novo domestication was also employed in a wild rice relative with the aim of developing the first ever domesticated polyploid rice crop [165]. After optimizing an efficient transformation system and generating a high-quality genome assembly for the allotetraploid wild rice *Oryza alta*, homologs of seven domestication-related genes and 113 agronomically important genes from diploid rice were edited using a CRISPR-Cas9 multiplex approach [165]. In this way, desirable characteristics for multiple agronomically important rice traits were obtained, thus representing a promising approach to address the challenges of global food security.

Despite the promising potential of rapidly domesticating new crops from their wild relatives, there are several bottlenecks within the de novo domestication pipeline that need to be considered [166,167]. For example, working with CWRs inherently introduces challenges because many of their traits are not well suited for agronomic settings, such as large non-compact growth forms and a high affinity for seed/pod shattering [168]. This also extends to genetic transformation systems, as many CWRs are onerous to regenerate [169]. In addition, for the effective domestication of CWRs within a single generation and therefore short timeframes, a multiplex editing approach targeting at least several domestication genes is likely required [167]. This, as well as establishing genomic resources for CWRs (i.e., annotated genome assemblies), can be a laborious and costly processes. Advances in genome editing technology will undoubtably aid in overcoming these bottlenecks in the coming years. For example, a more recent system to deliver genome editing reagents utilizes viral vectors, which completely bypasses the need for regeneration [170]. De novo domestication represents a novel pathway toward developing new crop species with diverse gene pools. While there have been several successful examples that serve as guides, progress in domesticating CWRs will largely be governed by the rate of characterization of domestication genes in major crops and the identification of agronomically beneficial traits in CWRs [171].

### 4.2. Gene Stacking in Polyploid Crops

Crop wild relatives and landraces represent a natural reservoir of diverse alleles, many of which are agronomically favourable. However, such natural alleles are restricted to certain species or closely related species and are often not exploited for other cross-genus crops. Even within its own closely related species, there is a chance that alleles cannot be introgressed effectively, due to genetic and genomic barriers such as linkage drag, which may introduce undesirable genes; genome incompatibility, resulting in unsuccessful fertilization; and differences in genetic background, causing a lack of gene expression, among others [172,173,174]. One of the ways to circumvent these introgression problems and allow alleles to be transferable to other crop systems in a more controlled manner is to generate alleles synthetically using genome editing. One successful example is the cloning of the candidate gene *FT-D1* in wheat, which is associated with total spikelet number and heading date, where the allele is conserved in hexaploid wheat and is suitable to be transferred across wheat species with different genetic backgrounds, and could potentially be pyramided with other genes controlling yield to improve wheat yield [175] (Figure 3B).

CRISPR multiplex editing offers great promise in accelerating the crop breeding process, where multiple gRNAs can be designed to target sequence modification at multiple genomic sites. Gene pyramiding using a multiplexing strategy in an elite winter wheat variety, Zhengmai 7698 (ZM), was recently performed using six constructs of gRNA targeting three homeologs for each gene with a tandem array of 2, 3, 4 and 5 tRNA-gRNAs each in a single transcript unit targeting 6, 9, 12 and 15 genomic loci in common wheat, which translates into a shortened breeding cycle time to reach the desired phenotypic outcomes, from 10 years to within 1 year [176]. In *Brassica oleracea*, multiple gene edits targeting the self-incompatibility, male sterility and phytoene desaturase genes were implemented to achieve male sterility for breeding pure F1 hybrid cabbage seeds in a more efficient way than the conventional backcrossing strategy [177]. Based on these achievements, we are now able to study and better understand the regulation of paralogous genes in polyploid crops, which increases the success rate of gene stacking efforts in these highly complex genomes. Efficient delivery systems of gRNA into the host plant and its subsequent transgene expression play very important roles in multiplex genome editing [178]. Plant-based virus vectors such as Beet necrotic yellow vein virus (BNYVV) and Potato virus X have been reported as a highly efficient tool to deliver the gRNA in multiplex forms into *N. benthamiana* plants, with up to four simultaneous expressions of recombinant proteins within the same cell, by making use of the virus replication method to accumulate large amounts of gRNA with an advantage of having a transgenerational effect [179,180].

### 4.3. CRISPR Screen

Genome-wide CRISPR screening is a powerful approach to characterize genes at the genome level and discover gene–phenotype relationships in various biological systems. Genome scale screening of mutations, deletions, transcriptional activation or repression in population genomics studies, or any fundamental biology studies, can be performed with CRISPR gRNA libraries, which allows for high-throughput applications in functional genomics studies of plants [181] (Figure 3C). An extension of this approach includes pooled CRISPR screens, where a library of cells, each receiving different gRNAs, is created and these gene-edited cells are subjected to selective pressures to induce the cell’s competitiveness, and with those cells that are fit, their encoded gRNAs are read out via high-throughput sequencing [182]. Pooled CRISPR can be linked with single-cell transcriptomes in CRISPR droplet sequencing (CROP-seq) such that the effect of the edit on thousands of heterogenous individual cells can be studied [183,184]. Other similar technologies such as CRISPR-seq and Perturb-seq are examples of how CRISPR is linked with single-cell transcriptomes to study the effect of multiple gene disruptions on gene transcription at the single-cell level in a high-throughput manner [185,186,187]. The latest advancement in CRISPR screening is the coupling with imaging technologies that allow us to visualize the changes in the cellular phenotypes incurred by the edited gene [182], for example locating the long non-coding RNA in the nuclear compartment [188]. In crop breeding, a pooled CRISPR screen using a small-scale CRISPR library (5–10 gRNA) with multiple alleles per gRNA was implemented in soybean, targeting 102 candidate genes in 407 transgenic lines, thus achieving multiplex mutations within the genome of one transgenic plant [189].

### 4.4. Gene Drive

Gene drive is the process in which the allele of a gene is converted into the desired version through deliberate means, where it can be the CRISPR-Cas-based approach, and the outcome is transforming heterozygote cells into homozygotes so that all the offspring carry the desired allele at 100% frequency [190]. The gene drive technology using CRISPR has proved successful in the malaria mosquito vector, *Anopheles gambiae*, targeting the inactivation of the gene controlling female reproduction with a progeny transmission rate of 91.4 to 99.6% [191]. Another example was also reported on an insect, *Drosophila suzukii*, a type of crop pest [192]. Successful allele sequence replacement in gene drive systems relies on efficient homology-directed repair, HDR, with a high transmission rate, but this is often lacking in plants [193,194]. However, it was shown that higher efficiency of HDR can be achieved in *Arabidopsis* during early egg cells and/or early embryos stages and immediately after transformation before T-DNA integration [195]. With improvements in the efficiency of the HDR mechanism in plants, the application of the CRISPR gene drive in plant breeding is promising and will gain momentum in future speed-breeding projects. For example, to accelerate the breeding of pathogen-resistant crop varieties possessing the mutated susceptibility (*S*) gene (*S*-mutants), the CRISPR gene drive system can be used to develop a homozygous *S*-mutant in a single parental line and re-use this parental line in numerous crossing experiments to produce high-quality cultivars without the need to perform CRISPR edits on each parent for every hybrid combination, thus saving time and resources [196]. Other potential applications of the CRISPR gene drive in plants include weed control [197], improvement of yield traits [198] and homoeologous allele editing in polyploid crop species with several factors to be considered for effective gene drives, such as the plant life history, the DNA repair mechanism, the potential for unintended evolutionary responses and ecological risk [194].

### 4.5. Application of CRISPR-Cas9 in Addressing Food Security Issues

Considering the compounding global shocks, such as the COVID-19 pandemic, the Russia–Ukraine conflict and climate changes, which have impacted food security in recent years, there is immense pressure to improve crop breeding. The production of hybrid seeds could greatly facilitate stable crop yield and ensure seed quality. For example, the application of male sterility technology in hybrid maize has benefited the smallholder farmers and breeders in Africa by increasing the yields and reducing seed production costs [199]. To expedite the lengthy process of producing male sterile lines for hybrid breeding, CRISPR-Cas9 has been employed in rice [200], wheat (*Triticum aestivum*) [201] and soybean (*Glycine max*) [202], thereby expanding the germplasm breeding pool for crop heterosis. Genome editing tools have also been utilized in studies related to crop nutrient uptake, such as the gene functional characterization of the *OsZIP9* gene in rice, which plays a role in Zn uptake in soil [203], and the *ZmbHLH121* gene in maize, a transcription factor that regulates root cortical aerenchyma, enhancing uptake of water and nutrients from the soil [204]. Successful applications of CRISPR/Cas9 systems in these areas have clearly demonstrated the potential to improve crop adaptability to marginalized soils, a recurrent problem in agricultural lands in developing or poor countries. Additionally, crop breeding can be further accelerated through Haploid-Inducer Mediated Genome Editing (IMGE), as demonstrated in maize plants that have been edited to produce homozygous pure doubled haploid (DH) lines with the desired trait improvement, carrying the elite maize inbred line genetic background within two generations [205], thus expanding the crop breeding tools to secure food for the future.

## 5. Challenges, Prospects and Conclusions

As naturally evolved immune systems identified in bacteria and archaea, CRISPR-Cas systems present inherent programmable gene editing capacity, which facilitates genome engineering and leads to revolutionary findings in biological sciences. In this review, we delineate the three genome editing phases using CRISPR/Cas systems, summarize the distinct structural and functional features of the two classes, consisting of three types each, and exemplify the applications in the plant genome editing of typical effectors.

One of the major bottlenecks hindering the widespread application of plant genome editing is the arduous tissue culture protocol required to create gene-edited plantlets. This step is not only labour- and time-intensive, but also presents drawbacks such as species-dependent feasibility. Recently, the adaptation of the de novo meristems induction method brings glimmers of light to overcome this barrier, via inducing the differentiated cells to form meristems. Gene editing components, such as Cas proteins and gRNA, are combined with specific regulation effectors to be delivered into seedlings or somatic cells of soil-grown plants outside aseptic conditions. Gene-edited shoots are induced, which eventually grow into gene-edited plants [156]. Once the existing limitations of this method are overcome, it will advance plant genome editing with faster editing protocols on an expanded range of plants.

The other limitation of CRISPR systems is in regard to the delivery of plasmid DNA, which often suffers from low transformation efficiencies, species-dependence and transgene integration into the host genome [206]. However, these limitations have now been addressed by using nanoparticles such as carbon nanotubes carriers. In this approach, DNA is adsorbed onto the surface of the nanoparticle, which is then successfully delivered into monocots, such as rice and wheat, and dicot plant species, such as tobacco, arugula (*Eruca sativa*) and cotton, without integration into the plant nuclear genome. It is worth noting that the large size of Cas9 requires optimization steps to achieve better adherence of the plasmids onto nanoparticles [206,207,208]. Another delivery mechanism, cell-penetrating peptides, make use of the peptide’s properties to effectively transport various biomolecules such as sgRNA and RNPs to its targeted location within the cell, through interactions with the cell membrane [209]. This approach is ideal for the cellular uptake of large biomolecule editing components into target cells and has been shown to be effective in a wide range of cell types [210].

Another major setback of implementing the CRISPR-Cas system in the plant science commercial sector relates to the complexity surrounding patent licensing and commercialization, as no single entity claims ownership in all aspects of the CRISPR-Cas9 technology. According to patent analytics data from IPStudies, a Swiss-based IP management consultancy, there were more than 11,000 patent families applications filed for technologies on different aspects of CRISPR-Cas9 systems such as gRNA-guide sequence, nuclease, vector delivery, CRISPR sequence, among many others [211]. The CRISPR patent grant holders enjoy a 20-year monopoly on the technology with the requirement of making the details of the work transparent to all researchers [212]. The main patent holders—the Broad Institute and the University of California—have granted certain “surrogate” companies the license, whereby the companies have the right to commercialize the CRISPR product and gain all the other perks of a patent owner, including sublicensing it out to other commercial enterprise [213]. For example, in plant science, Monsanto and Bayer Crop Science obtained a non-exclusive license from the Broad Institute (Harvard and MIT), while Dupont Pioneer acquired an exclusive license from the University of California-Berkeley and University of Vienna through the “surrogate” Caribou Biosciences, and later entered into a joint non-exclusive licensing agreement with the Broad Institute for commercializing and developing CRISPR-Cas9 products [214,215,216]. In plant breeding applications, Corteva Agriscience (Agricultural division of DowDuPont) and The Broad Institute of Harvard and MIT have licensed the CRISPR/Cas 9 technology to one of the largest agriculture companies in the world, the J.R. Simplot Company, which has successfully marketed the gene-edited potato varieties that are resistant against pathogens with reduced acrylamide, and is now working to edit the genes of strawberries for better qualities, in partnership with Plant Sciences, Inc [217,218]. These for-profit patent ownership of “surrogate” setups have obvious drawbacks, the main one being a limitation of commercial development by small biotechnology companies who may wish to apply CRISPR technology in their R&D innovation, thus hindering application-oriented improvement of CRISPR technology [219].

The exorbitant licensing fees for CRISPR technology which can reach USD 100 million have hindered many plant agricultural commercial companies from using this technology [220]. As a workaround strategy, these companies could work on different enzymes and systems that fall outside of the CRISPR-Cas9 patents. Research universities that are patent holders should offer free licenses to non-profit organizations, exemplified by Wageningen University and Research providing free licenses to non-profit entities for using CRISPR technology in their effort towards food security [221]. Additionally, there has been a huge call for making the CRISPR license free of charge for research institutions while giving the patent holders claim rights of any invention coming out from these institutions [222].

CRISPR-Cas systems are considered as a potent tool in synthetic plant biology, especially in synthetic plant genomes. The versatility of CRISPR-Cas genome editing tools may make plant artificial sequences no longer a fantasy, via manipulating and designing gene expression, transcriptional regulation and post-transcriptional behaviours using natural or artificial DNA-binding domains and nucleases [65,223]. To achieve this goal, CRISPR-Cas multiplex systems are under development, improving from a single catalytic function at multiple sites simultaneously [224,225], to multiple enzyme activities per site at multiple sites at the same time [223,226]. Gene stacking to add new and desirable traits by site-specific recombination on chromosomes has been realized, as reviewed above. Further endeavours are requested on the way to generate genuine plant artificial chromosomes (PACs) and finally synthetic plant genomes (Figure 3D).

This review highlights the various CRISPR subtypes containing unincluded effectors, which show distinct structural and functional differences from the currently applied CRISPR-Cas systems. These Cas proteins have been applied in mammalian cells or for human disease diagnosis and they offer great resources to expand the plant genome editing toolkit.

In conclusion, CRISPR-Cas systems have improved rapidly after their first identification as genomic engineering tools. The domains and nuclease activity in Cas proteins, gRNA components, plant species, transformation procedures and other details in the CRISPR-Cas systems all play an important role in determining precise genome editing. During the process, drawbacks of currently applied CRISPR-Cas systems, such as limitations in target recognition and collateral cleavage, have repeatedly been recognized and tackled unceasingly with now the availability of expanded CRISPR-Cas tools with higher editing efficiency, lower off-target activity and wider application. We believe CRISPR-Cas systems will become even more effective and flexible in plant genome editing to achieve global food security in the changing climate.

## Figures and Tables

**Figure 1 ijms-24-08623-f001:**
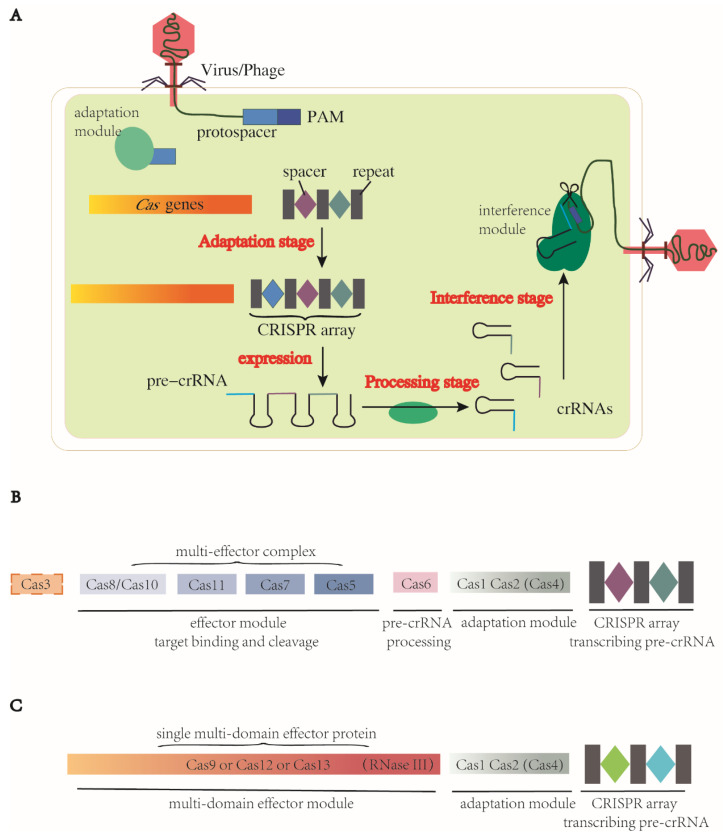
Three stages of genome editing via CRISPR-Cas systems and representative components of two classes of CRISPR-Cas system. CRISPRs encode pre-crRNAs. (**A**) CRISPR-Cas systems function in three stages. (1) Adaptation: spacers are acquired from foreign MGEs and inserted into CRISPR loci between repeats as immune memory. (2) Pre-crRNA expression and processing stage: CRISPR loci are transcribed into pre-crRNAs and processed into mature specific crRNAs. (3) Cas proteins and gRNA form a complex, recognize the homologous foreign sequence and cleave at target sites. (**B**) Class 1 CRISPR-Cas systems. Effector module consists of a complex series of Cas proteins. Protein Cas6 participates in the pre-crRNA processing. (**C**) Class 2 CRISPR-Cas systems. Their effector module is always a single protein with multi domains, including the function of pre-crRNA processing. In CRISPR-Cas9 systems, this function is realized by RNase III. Protein Cas1, Cas2 and sometimes Cas4 form a common adaptation module in both classes. These two figures present simplified architectures of two classes, with subtle difference and unknown domains between subtypes unshown here.

**Figure 2 ijms-24-08623-f002:**
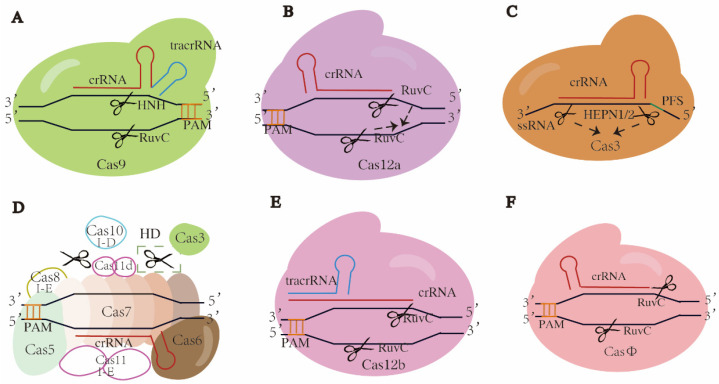
Schematics of typical or promising CRISPR-Cas systems. Cas proteins and crRNAs are targeted at DNAs except in C, which is single-strand RNA. Target sequence recognition is directed by PAM, except in C, by PFS. (**A**) Class 2 type II CRISPR-Cas9. A Cas9 protein forms a complex with crRNA and corresponding tracrRNA, cleaving dsDNAs at target sites and producing blunt ends. (**B**) Class II type V CRISPR-Cas12a. Single Cas12a protein forms a complex crRNA and cleaves dsDNA at target site, producing sticky ends. When activated, two RuvC domains would be superimposed and generate ssDNA activity, indicated by the dotted arrows. (**C**) Class 2 type VI CRISPR-Cas13. Single Cas13 protein forms a complex with crRNA and cleaves single-strand RNA of target gene. When activated, two HEPN domains would come close to each other and generate an active site, indicated by the dotted arrows. (**D**) Class 1 type I-D CRISPR-Cas10 or type I-E CRISPR-Cas3. In CRISPR-Cas10, Cas10 and Cas3 bind to the Cascade complex containing Cas5, Cas6, Cas7 and crRNA, then Cas10 cleaves and digests dsDNA at target site bidirectionally, indicated by the scissor in dotted box; HD is histidine aspartate. In CRISPR-Cas3, Cas3 is recruited to the Cascade complex containing Cas5, Cas6, Cas7, Cas8, cas11 and crRNA, and digests dsDNA at target sites. All filled patterns are common components shared by both subtypes, while the unfilled patterns are components specific to subtype as noted. The Cas3 in type I-D lacks the nuclease domain. Origins of Cas11 subunits differ between two subtypes, from independent ORF and Cas10d domains, respectively. (**E**) Class 2 type V CRISPR-Cas12b. Single Cas12b protein forms a complex with crRNA annealing to tracrRNA and cleaves dsDNA at target site, producing sticky ends. (**F**) Class 2 type V hypercompact CRISPR-CasΦ. Single CasΦ protein forms a complex with crRNA and cleaves dsDNA at target site, producing sticky ends. The scissors refer to where strand was cleaved and blunt or cohesive ends appeared.

**Figure 3 ijms-24-08623-f003:**
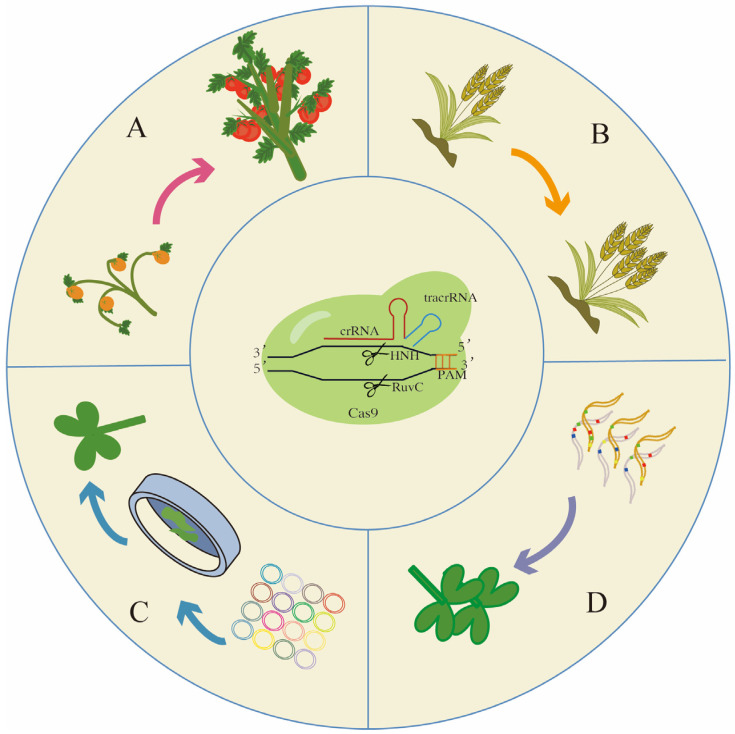
Further application of CRISPR/Cas systems in agricultural science: (**A**) De novo domestication demonstrated improvements in both plant architecture and yield. (**B**) Gene stacking of homeologs in polyploid crops can lead to improved yield traits such as spikelet number. (**C**) The prospective application of CRISPR screening using gRNA libraries. (**D**) Its potential application in producing hybrid vigour crops for future food security. Arrows represent symbols for gene editing.

**Table 1 ijms-24-08623-t001:** Characteristics of the six types in CRISPR-Cas systems.

Class	Type	Substrate	gRNA	Nuclease Domains	Stages and Related Cas Proteins				Indels
					Adaptation	crRNA Processing	Target Binding	Target Cleavage	
1	I	dsDNA; ssDNA	crRNA	HD	Cas1, Cas2, Cas4	Cas6	Cas5, Cas7, Cas8, Cas11/Cas5, Cas7	Cas3/Cas10	Long-range deletions
	III	RNA: DNA	crRNA	RRM	Cas1, Cas2	Cas6	Cas5, Cas7, Cas10, Cas11	Cas10	Degraded RNA/RNA
	IV *	DNA	crRNA	HD/lost	Cas1, Cas2/lost	Cas6	Cas5, Cas7, Cas11, Csf1	lost	Unknown
2	II	dsDNA	crRNA, tracrRNA	HNH, RuvC	Cas1, Cas2, Cas4	Cas9, RNase III	Cas9	Cas9	Small indels
	V	dsDNA; ssDNA	crRNA/crRNA, tracrRNA	RuvC	Cas1, Cas2, Cas4	Cas12	Cas12	Cas12	Small indels
	VI	RNA	crRNA	HEPN	Cas1, Cas2	Cas13	Cas13	Cas13	Degraded RNA

* Some type IV CRISPR-Cas systems lack adaptation module, Csf1 and/or Cas6; some information remains unknown.

**Table 2 ijms-24-08623-t002:** The advantages and limitations with suggested solutions applicable to the different CRISPR systems.

CRISPR Systems	Advantages	Limitations	Solutions
CRISPR/Cas9 [61,155,156]	SpCas9 system is the first CRISPR/Cas system to be used in genome editing and shows robust activity.	Its specific requirement for NGG PAM hiders its wider application in precise genome editing. The large size of Cas9 hinders its editing efficiency and packaging load. It introduces off-target mutations. Its transformation systems rely on tissue culture, which is labour-intensive.	The identification of other Cas9 orthologs expands the variety of recognized PAM sequences. gRNA structure could be optimized to increase its editing efficiency. High-fidelity Cas9 varieties can be designed; nCas9 and dCas9 lead to increased specificity; highly specific gRNA could be designed. Tissue-culture-free skills have been developed to simplify the process. Development of other CRISPR systems, such as CRISPR/Cas12a, could overcome some major limitations using CRISPR/Cas9 system.
CRISPR/Cas12a [157,158]	It requires only a crRNA and can process pre-crRNA into mature crRNA itself. It has high fidelity.	It only recognizes TTTV PAM. Its engineering is temperature-sensitive in plants.	Variants have been engineered to recognize alternative PAM sequences. Cas12a variety with high activity at lower temperature has been identified.
CRISPR/Cas13 [94]	It is the only known CRISPR/Cas system to bind and cleave RNA. It has a small size to be packaged into a virus-based vector.	It may result in collateral nonspecific RNA degradation.	Engineered dCas13 with inactivated catalytic domain that is fused with RNase domain can increase precision in RNA targeting.
Type I subtypes [115,122]	The Cascade structures exhibit higher specificity by recognizing longer target sequences than Cas9 systems. It can create long deletions.	Its wider application is inhibited by the multi-subunit effectors.	Each subunit could be modified to achieve various engineering purposes.
CRISPR/Cas12b [134,140]	It induces deletions longer than those induced by Cas9 system.	It requires high temperature to cleave DNA.	Protein engineering has been conducted to reduce its optimal cleavage temperature and it could be properly applied in genome editing of heat-tolerant plants.
CRISPR/CasΦ [142]	It has a hypercompact structure and is capable of being packaged in virus-based vectors. It recognizes a wide range of PAM sequences.	The size of packaged DNA is limited when using a virus-based vector.	This system is at a quite early study stage and further investigation will expand its application in the future.

## Data Availability

Not applicable.

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
