# Peer review of "CRISPR-Based Genome Editing Tools: An Accelerator in Crop Breeding for a Changing Future"

_ijms, 2023, doi:10.3390/ijms24108623_

Round 1
Reviewer 1 Report
The manuscript by Fangning Zhang et al described the development of CRISPR-Cas systems, outlined their classification and features, reported their natural mechanisms in plant genome editing, and characterized the applications in plant research. This manuscript is well written and organized. Recently, many papers about base-editing technologies to enhance plant production came out. If the authors could add some paragraphs about single base editing for plant genome editing, I think this paper will be better.
Reviewer 2 Report
Comments for the Authors
In the manuscript, the authors have given an exhaustive account of “Plant genome editing tools: an accelerator in crop breeding for a changing future.” The authors have put in a reasonable amount of effort, and the work is appreciable. This paper deals with an essential and exciting topic. The authors present their data in a novel and coherent approach, making it a compelling subject for plant breeders. The methods used are appropriate, and the conclusions are well-supported by the data. The paper adopts new research techniques and contributes to this field. I recommend an article to accept for publication after minor revision. For better readability of the manuscript, I would like to suggest some changes and improvements.
1. First, mention the complete name and later use an abbreviation, e.g., CRISPR/Cas9, etc. Names of the genes and plant names should be in italics; the first time, mention a complete name and later use an abbreviation for plant and any other organism.
2. There are several mistakes in the whole manuscript; please add or remove extra characters, and check again.
3. Some recent works about CRISPR/Cas9, as follows, should be considered for introduction and must be cited.
a. SU Khan, et al 2022, DOI: https://doi.org/10.1016/j.plantsci.2022.111435
b. SU Khan et al 2020, https://www.mdpi.com/1422-0067/21/16/5665
c. SU Khan et al 2022, https://www.mdpi.com/1422-0067/23/19/11156
4. Please ensure that all the units are in the same format.
5. English should be carefully checked and improved. The paper suffers from poor grammatical sentences and mistakes.

Reviewer 3 Report
Reviewer’s comments
Comments:
1. In the manuscript entitled “Plant genome editing tools: an accelerator in crop breeding for a changing future,” the authors reviewed two classes (class 1 and 2) of the CRISPR/Cas systems and mechanisms of various Cas proteins, including Cas 12 and Cas13 proteins discovered and applications of various Cas proteins to improve crop quality and yield. The authors did summarize the field (more detailed welcome) but did not write for a broad audience.
2. There are several serious weak points in this review. First of all, it needs to be better balanced. Some paragraphs and sentences are repeated throughout the manuscript. For example, “L606 to L616” the same sentences are repeated in L659 to L669.
3. L61 subsection is familiar and quite general for the CRISPR audience. Therefore, it can be removed from the manuscript.
4. The discussion of CRISPR/Cas type’s application in plant genome editing is weak. Results from only one or two studies are reported, which lack in-depth explanation or discussion of each Cas systems application in crop breeding (L138-this subsection should be extensively reviewed because of its wide applications).
5. L138-Subsection describing results must include gene targeted, vector name, and significance of targeting the genes. Summarizing the results in a table also can be helpful for the readers.
6. The advantages and disadvantages of each system on the application in crop breeding should be summarized.
7. The future applications of CRISPR/Cas systems in plants need to be discussed and laid out. Therefore, it needs better justification. A figure showing how CRISPR/Cas systems help in the crop breeding process in the future would go a long way in engaging the readers. This is a critical section which needs much improvements for insights, inroads of each tool for future crop breeding. For example, some tools are projected to improve the nutrient transport in other reviews (https://doi.org/https://doi.org/10.1016/j.biotechadv.2022.107963; https://doi.org/10.3389/fgene.2022.900897).
8. Because of the diverse nature of the review, the limitations are treated in a distal sense. The limitations of each Cas system covered in single sentences are completely insufficient.
9. Add some tables to specifically point out the limitations and offer some suggestions as to how these limitations can be overcome or at least challenged. That would make an interesting paper, not without.
10. Overall, in this review, several sections are wrapped up with a brief summary that more work is needed to improve the manuscript, and hence it cannot be recommended for publication in its current form.
Reviewer 4 Report
Presented review titled Plant genome editing tools: an accelerator in crop breeding for a changing future summarizes the development of CRISPR-Cas systems, outlines their classification and the applications in plant research.
The review is well written elaborating more on the CRISPR-Cas-based system, and it’s worth the authors considering a title that reflects the contents predominantly covered in the review rather than a generic title.
Round 2
Reviewer 3 Report
Reviewer’s comments
Comments:
Thanks for addressing the various questions and comments. There are, however, several unresolved and new issues with this revised manuscript. Especially the insights prion is very weak. Looks like a mere compilation of past results without their own insights. They were also asked to refer to a few articles which were not utilized to improve the article.
1. figure 3, on applications of CRISPR/Cas system in agricultural sciences, look very simple and without insights. So authors should expand the figure by including details like the development of male sterile lines, how to fix the hybrid vigor, haploid induction for breeding programs, how de nova domestication via CRISPR/Cas system can help the world to achieve SDG2, Chromosomal translocations, improving nutrient transport, etc.
2. The authors also fail to address the limitations of the delivery of CRISPR/Cas reagents, which is the primary obstacle for plant genome editing via CRISPR/Cas systems. Nowadays Nano materials like carbon nanotubes, DNA nanostructures, and cell-penetrating peptides are widely used as a promising delivery for CRISPR reagents. Therefore, the authors must update these details in the manuscript. Thera re some recent articles which should be referred to and cited here (eg. https://doi.org/10.1016/j.plaphy.2023.02.030)
Author Response
Thanks for addressing the various questions and comments. There are, however, several unresolved and new issues with this revised manuscript. Especially the insights prion is very weak. Looks like a mere compilation of past results without their own insights. They were also asked to refer to a few articles which were not utilized to improve the article.
Thank you for the feedback.
The articles cited in the manuscript were used to support the discussion points. We provided our insights, for example here:
Line 223-227: "For example, completely sterile spo11-1 mutants were generated in hexaploid wheat for the first time, using CRISPR-Cas9 to edit all three homoeologues of SPO11-1 in a loss-of-function manner [66]. This study not only revealed the role of SPO11-1 in meiosis, but also supported the possibility of editing meiotic recombination in wheat using CRISPR-Cas9 systems."
- figure 3, on applications of CRISPR/Cas system in agricultural sciences, look very simple and without insights. So authors should expand the figure by including details like the development of male sterile lines, how to fix the hybrid vigor, haploid induction for breeding programs, how de nova domestication via CRISPR/Cas system can help the world to achieve SDG2, Chromosomal translocations, improving nutrient transport, etc.
The suggestion here indicates that we need to discuss these ideas in the main text. We think that the current content is satisfactory for a review paper, and any additional text will make the paper excessively lengthy and cumbersome to read.
- The authors also fail to address the limitations of the delivery of CRISPR/Cas reagents, which is the primary obstacle for plant genome editing via CRISPR/Cas systems. Nowadays Nano materials like carbon nanotubes, DNA nanostructures, and cell-penetrating peptides are widely used as a promising delivery for CRISPR reagents. Therefore, the authors must update these details in the manuscript. Thera re some recent articles which should be referred to and cited here (eg. https://doi.org/10.1016/j.plaphy.2023.02.030)
Same here
Round 3
Reviewer 3 Report
The insight part is still weak. They didn't attend comments nos 1 and 2.
Author Response
Thanks for addressing the various questions and comment-s. There are, however, several unresolved and new issues with this revised manuscript. Especially the insights prion is very weak. Looks like a mere compilation of past results without their own insights. They were also asked to refer to a few articles which were not utilized to improve the article.
Thank you for the feedback.
The articles cited in the manuscript were used to support the discussion points. We have included our insights throughout the manuscript, in:
Lines 246-249, 306-209, 312-314, 348-352, 357-369, 396-400, 463-465, 472-475
- figure 3, on applications of CRISPR/Cas system in agricultural sciences, look very simple and without insights. So authors should expand the figure by including details like the development of male sterile lines, how to fix the hybrid vigor, haploid induction for breeding programs, how de nova domestication via CRISPR/Cas system can help the world to achieve SDG2, Chromosomal translocations, improving nutrient transport, etc.
Thank you for the suggestion. We have added a section to discuss these ideas in section 4.5.
- The authors also fail to address the limitations of the delivery of CRISPR/Cas reagents, which is the primary obstacle for plant genome editing via CRISPR/Cas systems. Nowadays Nano materials like carbon nanotubes, DNA nanostructures, and cell-penetrating peptides are widely used as a promising delivery for CRISPR reagents. Therefore, the authors must update these details in the manuscript. Thera re some recent articles which should be referred to and cited here (eg. https://doi.org/10.1016/j.plaphy.2023.02.030)
Thank you for the suggestion. We have added a paragraph to address this in section 5.
Round 4
Reviewer 3 Report
This work can now be accepted for publication.
Author Response
Thank you.